# Perceived warmth and competence predict callback rates in meta-analyzed North American labor market experiments

**Marcos Gallo**[1], **Carina I. Hausladen**[1,2]*, **Ming Hsu**[3], **Adrianna C. Jenkins**[4], **Vaida Ona**[4], **Colin F. Camerer**[1,5]

1 Division of Humanities and Social Science, California Institute of Technology, Pasadena, CA, United States of America, 2 Computational Social Science, ETH Zurich, Zurich, Switzerland, 3 Haas School of Business, University of California, Berkeley, Berkeley, CA, United States of America, 4 Department of Psychology, University of Pennsylvania, Philadelphia, PA, United States of America, 5 Computational and Neural Systems, California Institute of Technology, Pasadena, CA, United States of America

☯ These authors contributed equally to this work.
* carinah@ethz.ch

**Data Availability Statement:** To promote transparency and further research, all data underlying our study are openly available at the following GitHub repository: \url{https://github.

## Abstract

Extensive literature probes labor market discrimination through correspondence studies in which researchers send pairs of resumes to employers, which are closely matched except for social signals such as gender or ethnicity. Upon perceiving these signals, individuals quickly activate associated stereotypes. The Stereotype Content Model (SCM; Fiske 2002) categorizes these stereotypes into two dimensions: warmth and competence. Our research integrates findings from correspondence studies with theories of social psychology, asking: Can discrimination between social groups, measured through employer callback disparities, be predicted by warmth and competence perceptions of social signals? We collect callback rates from 21 published correspondence studies, varying for 592 social signals. On those social signals, we collected warmth and competence perceptions from an independent group of online raters. We found that social perception predicts callback disparities for studies varying race and gender, which are indirectly signaled by names on these resumes. Yet, for studies adjusting other categories like sexuality and disability, the influence of social perception on callbacks is inconsistent. For instance, a more favorable perception of signals like parenthood does not consistently lead to increased callbacks, underscoring the necessity for further research. Our research offers pivotal strategies to address labor market discrimination in practice. Leveraging the warmth and competence framework allows for the predictive identification of bias against specific groups without extensive correspondence studies. By distilling hiring discrimination into these two dimensions, we not only facilitate the development of decision support systems for hiring managers but also equip computer scientists with a foundational framework for debiasing Large Language Models and other methods that are increasingly employed in hiring processes.

com/carinahausladen/SocialPerceptions-Predict-Callback}. There are no restrictions on data availability.

**Funding:** NSF DRMS grants 1851879 (ACJ), 1851745 (CFC), 1851902 (MH), and a Tianqiao and Chrissy Chen Graduate Fellowship (MG).

**Competing interests:** The authors have declared that no competing interests exist.

## Introduction

Discrimination is costly for organizations and detrimental to society. Recognizing discrimination goes beyond the moral and legal obligations—it speaks to the essence of strategic management, affecting team synergy, innovativeness, and overall organizational performance, areas central to modern management research and practice. Labor market discrimination occurs when individuals are treated unequally based on their observable characteristics, even when those characteristics should not impact expected job performance. Despite increasing awareness of the advantages of diverse teams among employers, and ongoing civil rights activism leading to legal protections against many types of identity-based discrimination, people from marginalized groups still face disparate treatment in the labor market [1]. However, because employer subjective expectations of productivity are rarely observed, it is difficult to conclusively pinpoint specific instances of discrimination [2, 3].

To try to control for subjective expectations, experimental "correspondence (or audit) studies" were developed starting in the 1960s [4–6]. Correspondence studies strive to control expectations by creating identical pairs of artificial resumes (with matching backgrounds, skills, education, etc.) and sending them to potential employers. Typically, only one categorical factor –such as race, gender, or sexuality– is varied between each matched resume pair. Everything else on the two resumes is the same. The test statistic is the difference in callbacks for the controlled variable. These studies have documented common patterns of discrimination across different social categories.

Despite their success in documenting discrimination based on single social identities, several important challenges remain. For instance, most studies fail to account for intersectionality of multiple social identities combined in complex and non-additive ways to influence treatment within the labor market [7–9]. People with multiple marginalized identities are subjected to more frequent and severe workplace harassment [10] and experience more obstacles to promotion [11].

To address this core limitation, we explore the extent to which stereotyped responses to social groups, as identified by correspondence studies, are associated with social perceptions of those groups. Perceptions are measured in a two-dimensional space of warmth and competence based on extensive evidence that the two-variable warmth-competence reduction robustly explains a surprising amount of variation across perceptions and behavioral reactions to social categories. *Warmth* is the perception of how good or bad another person's intentions are. *Competence* refers to how capable a person is of acting on their intentions [12].

Emerging research suggests that stereotypes about warmth and competence may contribute to labor market discrimination [6, 13–15]. In particular, in a recent analysis [13], applicants whose racial group was associated with higher perceived warmth received significantly more callbacks based on data from two field studies. Furthermore, when averaged across raters, warmth and competence scores for different groups are highly consistent across samples, suggesting that they reflect culturally shared stereotypes rather than idiosyncratic individual social perceptions [13]. Although suggestive, this evidence comes from a few studies in a limited set of hiring domains.

Our paper provides academic and practical contributions to the discourse on hiring discrimination. Academically, we show that social perceptions impact hiring decisions within studies that use names as a social signal. We broaden the scope by examining social signals beyond names, demonstrating their varied impact on hiring discrimination. Practically, we translate our findings into actionable strategies for hiring managers. Our framework also facilitates discrimination prediction, especially within under-researched, intersectionally

stereotyped groups. Critically, our research also proposes a framework for correcting biases in Large Language Models, whose application in recruitment is expanding rapidly [1].

Fig 1 provides an overview of the correspondence studies we use, classifying them according to the investigated categories and the number of "signals" used to convey these traits. Typically, an applicant's group affiliation is not explicitly stated but is subtly signaled through associations with 1) names indicative of race, gender, or age and 2) other characteristics (e.g., membership in a college LGBTQ club to signal sexual preference). Signals are chosen when possible to maximally distinguish groups (e.g., Sarah Davis (white, female), Deshawn Jefferson (black, male)). Note that, unlike studies using names, category studies usually employ a limited number of distinct signals; therefore, we choose to analyze name and category studies separately.

## Materials and methods

### Transparency and openness

We conform to standards of Preferred Reporting Items for Systematic Reviews and Meta-Analyses (PRISMA, [16]). All meta-analytic data, analysis code, and research materials (including our coding scheme) are available at (https://github.com/carinahausladen/NSF-Discrimination). Analysis was carried out with R version 4.2.2. Meta models were estimated with packages `metafor_3.8-1` and `meta_6.2-1` [17]. This review project was not preregistered. The study was not preregistered. Furthermore, the literature review for this study was based on North American studies included in [6], as well as records identified from Google Scholar and Web of Science. The date of the most recent search of the literature review conducted for the study was January 2023.

The study was approved under block approval IRB 0253, which was issued by the Caltech IRB board. Informed consent was obtained through Qualtrics before participants started the study.

### Data

One hundred ninety-one studies were gathered by combining those included in [7] and in our screening process (Fig S1 in S3 File shows a PRISMA diagram). For each study, we extracted information on the callback rates for each group, along with study-specific characteristics. Furthermore, we searched for published raw datasets for each study in the meta-analytic database. "Raw" means that data contain observations, including names or category signals and callback rates, for each resume sent in the experiments. We requested authors provide these raw data from their study for unpublished datasets. Table S1 in S3 File shows the datasets gathered. Additionally, Prolific participants provided ratings of warmth and competence for each category signal from the meta-analytic data and each name from the raw datasets.

### Statistical analysis

For meta-analytic analyses based on correlations, we deployed a random-effects model.

We chose a random-effects model for our analysis due to the inherent heterogeneity across studies. A random-effects model is suitable for our varied dataset as it accounts for differences between studies beyond sampling error, unlike a fixed-effects model. Additionally, since each study contributed a single effect size in our analysis of names, a mixed-effects model would not offer additional analytical benefits.

Each correlation $r$ is transformed into Fisher's z: $z = 0.5 \log_e \left( \frac{1+r}{1-r} \right)$, to ensure that the sampling distribution is approximately normal. The model is adjusted via the Hartung-Knapp

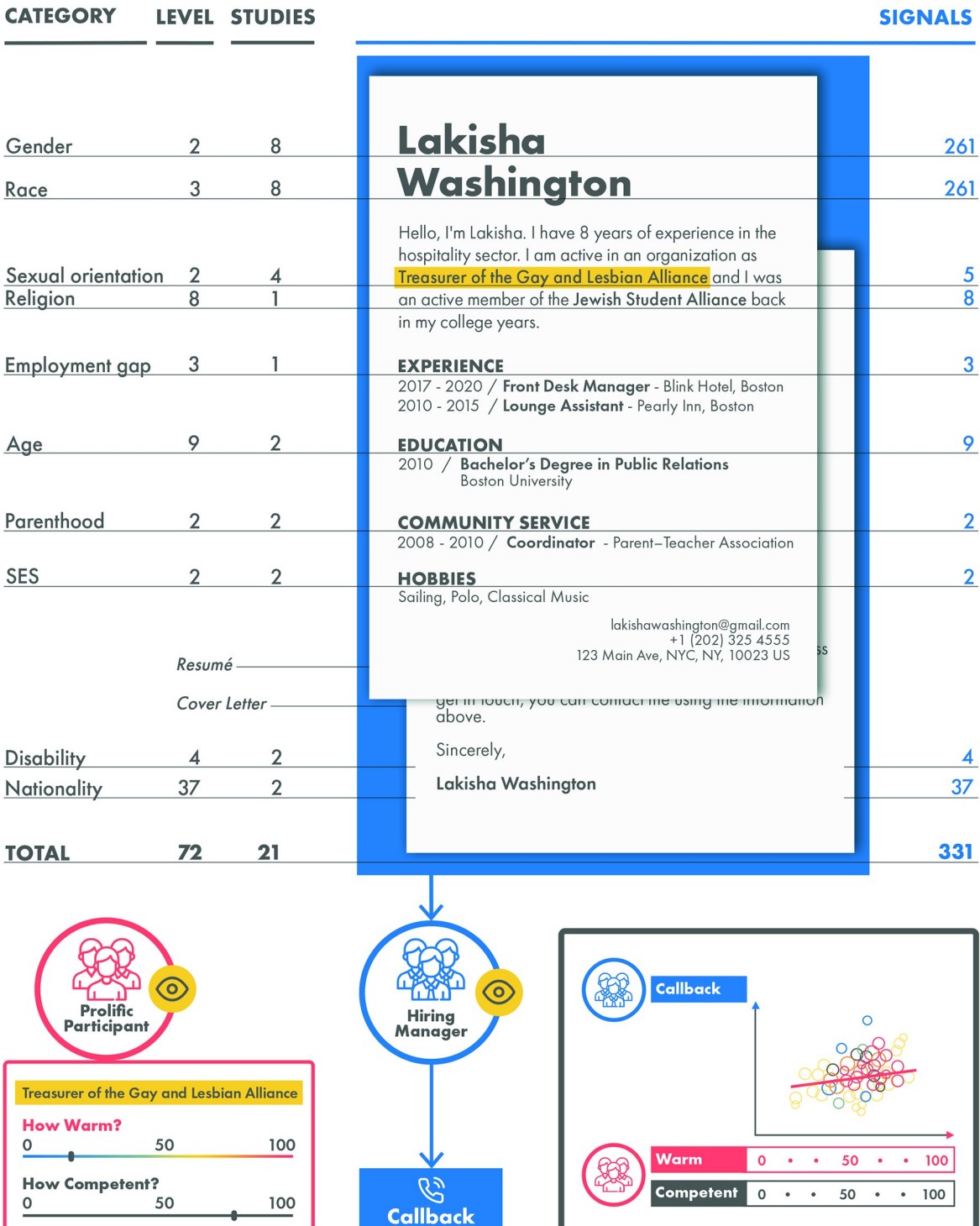

**Fig 1. The total number of studies, categories, and signals included in our meta-analysis, along with our statistical estimation strategy.** The numbers indicate the total counts of studies, categories, and signals. For a detailed overview of signals by category for which raw data was obtained, refer to Table S1 in S3 File. Please note that the total number of studies is 21, as some are included in more than one category. Example signals are presented in the middle column (the resume). The data sources are shown on the right-hand side: hiring managers made callback decisions based on resumes (in blue). Separately, we collected warmth and competence ratings on prolific, where participants (in red) only saw the respective signal (indicated in yellow). Our estimation strategy is visualized in the grey box in the bottom right corner: we used the averages of warmth and competence ratings to predict the callback percentage.

**Table 1. Linear probability regressions of callback rates on principal components and social perception ratings.**

|  | estimate | 95% CI | | p-value | SE |
|---|---|---|---|---|---|
|  |  | lower | upper |  |  |
| Meta regression for categories[1] |  |  |  |  |  |
| intercept | -0.32 | -1.06 | 0.43 | 0.08 | 0.37 |
| PC1 | 1.16 | -0.28 | 2.59 | 0.12 | 0.72 |
| PC2 | -0.62 | -3.58 | 2.35 | 0.69 | 1.49 |
| Meta regression for names[2] |  |  |  |  |  |
| intercept | -1.97 | -2.47 | -1.48 | 0.00 | 0.25 |
| PC1 | 1.00 | 0.41 | 1.58 | 0.00 | 0.30 |
| PC2 | 0.56 | -0.83 | 1.96 | 0.43 | 0.71 |
| Correlations $\rho$(callback, variable) for names[3] |  |  |  |  |  |
| PC1 | 0.33 | 0.03 | 0.66 | 0.03 | 0.13 |
| warmth | 0.34 | 0.08 | 0.64 | 0.02 | 0.12 |
| comp | 0.26 | -0.06 | 0.58 | 0.09 | 0.14 |

Note:

[1] Mixed-Effects Model (79 levels; $\tau^2$ estimator: ML)

[2] Mixed-Effects Model (691 names; $\tau^2$ estimator: ML)

[3] Three separate multivariate correlations; Random-Effects Model (8 studies; 725 observations); Inverse variance method, restricted maximum-likelihood estimator for $\tau^2$, Q-Profile method for the confidence interval of $\tau^2$ and $\tau$, Hartung-Knapp adjustment (df = 7), prediction interval based on t-distribution (df = 6), and Fisher's z transformation of correlations.

modification [18]. To estimate the random-effects model, the variance of the distribution of true effect sizes, $\tau^2$, has to be estimated, for which we deploy Maximum Likelihood [19]. The confidence intervals around $\tau^2$ are estimated via the *Q-Profile* method [20]. Furthermore, we calculate the $I^2$ statistic [21] to measure between-study heterogeneity.

Prediction intervals provide a valuable tool for estimating the likely range of effects that future studies may have based on the current evidence. As opposed to confidence intervals, prediction intervals consider $\tau^2$ to estimate the likely range of effects of future studies.

The meta-regressions (Table 1) were specified as mixed-effects models:

$\hat{\theta}_k = \theta + \beta x_k + \epsilon_k + \zeta_k$. The first error, $\epsilon_k$, represents the sampling error through which a study's effect size deviates from its true effect. The second error, $\zeta_k$, indicates that even the true effect size of a study is only sampled from an overarching distribution of effect sizes.

## Heterogeneity analysis

We visualized the contribution of each study to the overall heterogeneity against its influence on the pooled effect size (Baujat plot, Fig S2 in S3 File). We also computed several influence diagnostics (Externally Standardized Residuals, DFFITS Value, Cook's Distance, Covariance Ratio, Leave-One-Out $\tau^2$, Hat Value, Study Weight, Fig S3 in S3 File). A leave-one-out robustness analysis was used to point to the study whose exclusion results in the largest decrease in the $I^2$ statistic (Fig S4 in S3 File). Additionally, we implemented a Graphical display of heterogeneity (GOSH) plot analysis (Fig S5 in S3 File).

## Intraclass correlation (ICC)

We calculated the ICC through a two-way random-effects model (as provided by package `psych`) to assess the reliability of the average of $k$ ratings for each signal $i$. We describe each

rating as $y_{ij} = \mu + r_i + c_j + e_{ij}$, where $\mu$ is the average rating, $r_i \sim N(0, \sigma_r^2)$ and $c_j \sim N(0, \sigma_c^2)$ are random effects for the signals and raters, respectively, and $e_{ij}$ is the error term. Then, we compute ICC $= \frac{\sigma_r^2}{\sigma_r^2 + (\sigma_c^2 + \sigma_e^2)/k}$ [22].

## Finite mixture models (FMMs)

We used an FMM to generate two latent classes with distinct effects of PC1 on callback rates, with the probability of belonging to class $i$ defined as $\pi_i = \frac{\exp(\gamma_i)}{\sum_{j=1}^{g} \exp(\gamma_j)}$, where $\gamma_i$ is a function of job characteristics (Table S11 in S3 File).

## Demographic characteristics of prolific participants

We aim to predict callback rates based on perceived stereotypes. Recognizing that stereotypes are shaped by cultural contexts, we selected raters with North American backgrounds to match the cultural perspectives of recruiters in North American labor market experiments.

Participants were recruited through Prolific to provide warmth and competence ratings, with a total of 787 raters for both names (averaging 85.9 per name) and categories (averaging 99.1 per category level). The number of participants to recruit was guided by literature [13] and a point of stability estimation, indicating that mean ratings would stabilize with no significant changes beyond approximately 90 participants (detailed sample size information in S4 File.

Following the rating process, participants provided demographic information. In the group assessing names, 57.52% identified as female, with an average age of 37.62 years. The majority ethnicity was White/Caucasian (62.38%), and the prevalent educational attainment was a bachelor's degree (33.91%). In contrast, for the group evaluating categories, females comprised 50.1% of participants, with the largest age group being 25–34 years old (39.1%). This group also showed a higher proportion of White/Caucasian participants (77.7%), and 31.2% had achieved a bachelor's degree.

## Names: Social perception predicts callback in correspondence studies that vary names

We identified studies through a systematic search of correspondence experiments in North American labor markets (see PRISMA Flow Diagram, Fig S1 in S3 File). We further extracted name-specific callback rates from studies that reported or made them available through replication datasets for the following analyses. This procedure created a sample of eight studies.

Before examining warmth and competence, we first analyze how callback varies by race and gender. The difference between groups is summarized by the ratio of the callback rates of the potentially discriminated-against group compared to the benchmark group, with a ratio of 1 indicating perfect parity, ratios $< 1$ indicating negative discrimination, and those $> 1$ indicating privileged treatment.

In our sample, the callback ratio is $\hat{\theta} = 0.798$ for Black names, which was significantly less than one ($p = .07$). The same ratio computed by [6] is $\hat{\theta} = 0.68$ ($p < .001$). For the female gender compared to male, our estimated ratios are $\hat{\theta} = 1.02$ ($p = .36$) in the eight studies we have, agreeing with Lippens' $\hat{\theta} = 1.02$ ($p < .003$) [6]. Together the data show a 20–30% reduction in callbacks for Black names and no reduction for female names (Table S6 in S3 File). Our analysis did not differentiate between male and female-dominated occupations, which may account for the lack of a significant effect observed for females, as emphasized by [23].

To measure warmth and competence, lists of names from the correspondence studies were given to participants on Prolific (787 raters total, 85.9 per name). To evaluate the consistency of ratings across categories, we computed the intraclass correlation (ICC, as defined in Materials & Methods). Our results reveal that the level of agreement between raters differs across various studies, with agreement ranging from excellent to good in most studies (Fig 2A, Table S2 in S3 File). This variation is crucial, as low intraclass correlations of social categories create an upper bound on the reliability of the ratings (see Discussion for details).

The callback rates were computed by averaging the decisions of multiple hiring managers. Meanwhile, the warmth and competence scores were obtained from a different sample. To ensure reliable social perception measurements, we specifically recruited participants residing in North America with demographics closely resembling those of the average hiring manager, and we averaged ratings across raters. This enabled us to confidently match the social perception ratings with the callback rates per name.

Those warmth and competence ratings, across names in different studies, are shown in Fig 2A. There were only minor differences in warmth or competence ratings between black and white candidates or males and females (between 2 and 7 points on the 100-point scale).

Fig 2A shows strong, reliable positive associations between warmth and competence within all eight studies, ranging from .41 − .92 (Table S4 in S3 File). The pooled correlation is $\hat{\rho}_{w,c} = .78$ ($p < .001$). We, therefore, used principal component analysis (PCA). Fig 2D shows how the principal component scores (y-axis) are related to warmth and competence ratings (x-axis). The first component (PC1) reflects the positive association; it explains 79.3% of the variance. PC2 accounts for only 20.7% of the total variance, indicating its less prominent role in the overall data structure. Our subsequent analyses will focus on the PCs rather than the original warmth and competence ratings that generated them.

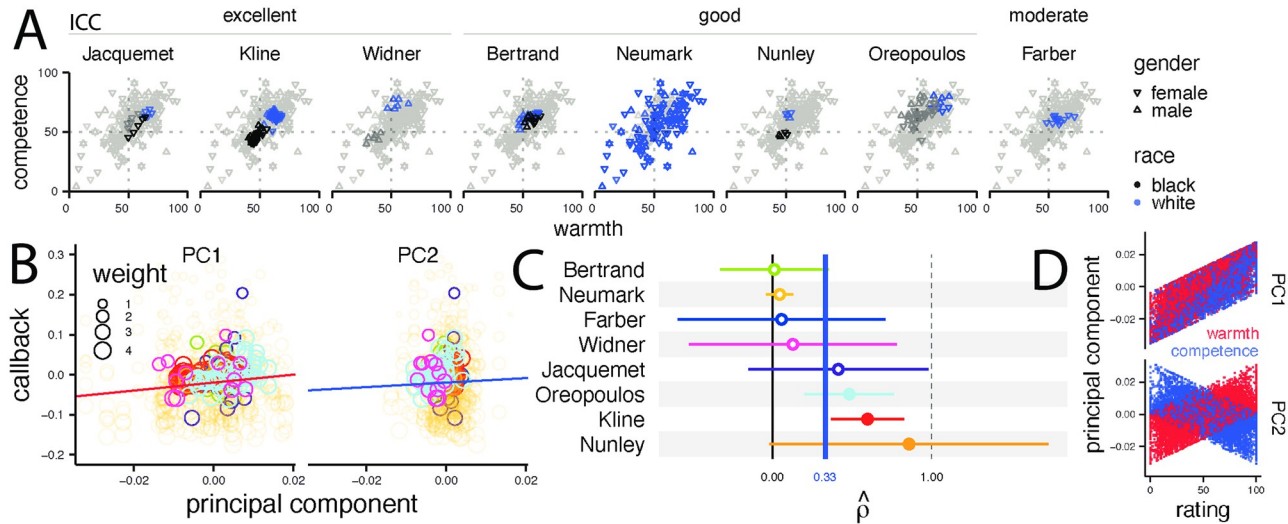

**Fig 2. Warmth and competence ratings across names and their association with callback rates.** (A) Each scatterplot shows warmth and competence for each name in the sample one study (with the first author name at the top). The correlations between the two rating scales are strongly positive in all eight studies (Table S4 in S3 File). (B) Correlations between callback rates and PC1 and PC2 components associated with specific names (aggregating all studies). Data from different studies are identified by colors, with the legend shown in panel (C). The slope coefficients, shown in Table 1 are $\hat{\beta}_{PC1} = 1.00(p < .001)$, $\hat{\beta}_{PC2} = .56(p = .43)$. (C) Forest plot of confidence intervals for study-specific estimates of the correlation between callback rate and the first principal component PC1, $\hat{\rho}(\text{callback, PC1})$. All correlations are positive. The pooled effect is $\hat{\rho} = .33$. (D) Scatter plots of name-specific warmth and competence ratings showing the structure of PC1 and PC2.

## Correlation between callback and PC1 as an effect

Fig 2C is a forest plot of the estimated correlations $\hat{\rho}$ (callback, PC1) and 95% confidence intervals of the eight studies.

The effects across studies were pooled via a meta-analytic random effects model. The pooled correlation between the callback percentage and PC1 is $\hat{\rho} = .33$ ($p = .03$), indicating a moderate correlation [24]. To interpret the pooled effect size meaningfully, we must consider the variance of the true effect sizes distribution, $\tau^2$, and the between-study heterogeneity, $I^2$ (see Materials and methods). As suggested by Fig 2C, there is "substantial heterogeneity" [25] among studies: 83 percent of the variation in effect sizes is due to between-study heterogeneity ($I^2 = .83$, 95% CI [.68−.91]). Furthermore, the variance of the true effect sizes distribution is significantly greater than zero ($\tau^2 = 0.08$, 95% CI [0.02−0.67]).

Given the large level of heterogeneity in our analysis, we find a wide prediction interval [26] (from −.40 to.81, details in Materials and methods), suggesting that future studies are likely to show a wide range of correlations, including negative ones. Therefore, caution is warranted in interpreting the results, and further research is needed to clarify the effect of social perception on callback.

In order to check the robustness of our findings and account for potential outliers, we conducted a comprehensive outlier and heterogeneity analysis. Only two out of eight tests (details in Supporting information) identified outliers which, when excluded, re-estimate $\hat{\rho}$ as.22 or.34. Both values remain close to the.33 all-study estimate in Fig 2C. Furthermore, Egger's regression test (Fig S6 in S3 File; intercept = 1.8, 95% CI [−0.25, 3.85], $t = 1.72$, $p = .14$) did not indicate publication bias.

As an alternative specification to the meta-analysis with $\hat{\rho}$, Table 1 reports results of a mixed effects model of callback rates against the PCs and raw ratings. The results are visualized in Fig 2B. The coefficient for PC1 is positive $\hat{\beta}_{\text{PC1}} = 1.00$ and highly significant ($p = .0008$).

The correlations for warmth ($\hat{\rho} = .34$; $p = .02$) and competence ($\hat{\rho} = .26$, $p = .09$) are similar to those observed for the first principal component (PC1). This small warmth-competence difference is consistent with much evidence that judgments of warmth are faster, more reliable, and more associated with behavior than competence judgments [27].

Moreover, we tested the predictive potential of our model for names. We found that $\hat{\rho}$ (callback, PC1) in [28] is closest to the pooled effect, and we, therefore, used data from this study to make predictions. Specifically, we trained a linear model on all names except one and then used this model to predict the callback for the left-out name. Fig S6 in S3 File visually represents our findings, with lighter shades of blue indicating lower callback rates. Our analysis reveals that callback rates are highest in the upper quadrant of the warmth and competence scale, while the callback rates are lowest in the lowest quadrant of this plot. Interestingly, we also observe clusters of the race that the names would signal (Black, White, or foreign-sounding).

## Exploratory analysis points to differential effects of social perceptions across job types

Previous studies suggest a link between the stereotype content of occupations and group affiliation [29, 30]. To investigate the role of social perceptions in determining callbacks across job types, we analyzed [31, 32], as those two studies provided adequate variation for meaningful conclusions. We employed finite mixture models (FMMs) to cluster occupations into two distinct groups for each study and examined the relationship between callbacks and PC1. Results reveal that the relationship between social perceptions and callbacks varies across job types. In [32], the correlations were slightly higher for advanced, specialized, or managerial positions

($r = .80$, $p = .009$ vs. $r = .77$, $p = .022$). For [31], the correlation between callbacks and PC1 was higher for service-oriented and less specialized jobs, though not statistically significant ($r = .26$, $p = .201$ vs. $r = .17$, $p = .340$).

An exploratory analysis using partial correlations indicated notable relationships between warmth, competence, and callbacks for different job categories. In [32], warmth was positively associated with callbacks in entry-level jobs but negative for higher cognitive and technical skill jobs ($r = .28$, $p = .320$ vs. $r = -.34$, $p = .200$). Competence, conversely, exhibited a positive association with callbacks, less pronounced in entry-level or sales-oriented roles ($r = .48$, $p = .213$ vs. $r = .83$, $p = .022$). [31] revealed a positive relationship between warmth and callbacks, stronger for jobs with greater social interaction requirements ($r = .35$, $p = .142$ vs. $r = .26$, $p = .188$), while competence displayed a negative association, more pronounced in professional or technical industries ($r = -.20$, $p = .281$ vs. $r = -.15$, $p = .302$).

## Categories: Mixed effects of social perception on callback rates in correspondence studies manipulating social identity categories

In the first section, we analyzed only correspondence studies that varied names. In the following, we investigate correspondence studies varying other categories, such as religious affiliation or membership in the LGBTQ community (Fig 1). Prolific participants (787 raters total, 99.1 by level) rated each signal on a scale from 0 to 100 within a category (e.g., how warm/competent they think a "treasurer in the gay and lesbian alliance" would be, Figs 1 and 3A). The intraclass correlation (ICC) [33] values vary across categories. Only two categories scored "poor",

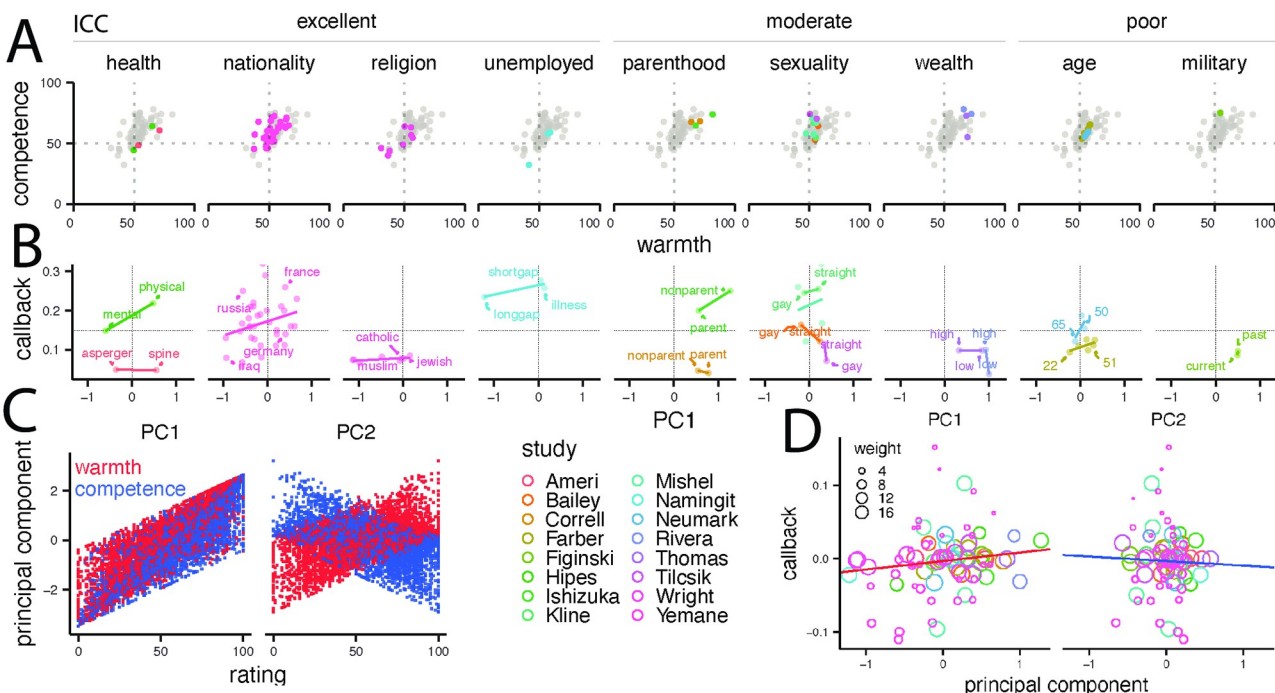

**Fig 3. Warmth and competence ratings across categories and their association with callback rates.** (A) Each scatterplot shows warmth and competence for each category-signal (with the category name at the top). The correlations between the two rating scales are strongly positive in all nine categories (Table S5 in S3 File). (B) Linear regression of PC1 on callback by category and study. Data from different studies are identified by colors, with the legend shown in the center of row three. (C) Scatter plots of category-specific warmth and competence ratings showing the structure of PC1 and PC2. (D) Meta-regression of PC1 and PC2 on callback, where each circle identifies a signal by study; the circle size indicates the assigned weight in the meta-regression. Lines indicate fitted intercepts and $\hat{\beta}$s.

while the remaining scored either "moderate" or "excellent" (Fig 3A). Note that, for sexuality and wealth, the different signals yielded vastly different ICCs, ranging from 0 to.83 (Table S3 in S3 File). These findings suggest that raters' agreement levels varied across those categories and signals.

Furthermore, we assessed the correlation between the warmth and competence ratings and found that the Pearson correlation index ($\rho$) was significant for most studies, with a pooled correlation of $\hat{\rho} = .595$ ($p < .001$, Table S5 in S3 File). The correlation of target warmth and competence ratings across studies makes it challenging to estimate the independent effects of these dimensions of social perception on the callback rates. To better capture the variability in social perception of different social categories, we conducted a PCA, revealing two principal components.

PC1 explained 80.73% of the variability in warmth and competence ratings, combining the positively correlated measures onto a single dimension. PC2 represented negative associations and accounted for 19.27% of variance. Therefore, we analyze the principal components rather than the ratings that generated them (Fig 3D).

In the following, we conduct a meta-regression pooling all studies to explore the extent to which the principal components predict callback. To increase the robustness of this analysis, we also perform a permutation test on our meta-regression models [21]. The resulting estimate for the coefficient of the first principal component is.01, which is not statistically significant ($p = .134$). Our model explains a small portion of the heterogeneity, accounting for only 3.31% (Table 1). Fig 3D visualizes the meta-regression model.

The meta-regression did not yield a significant overall effect. Therefore, we explored the relationship between PC1 and callback by category. Given the limited number of levels across categories (Fig 3B), it was not possible to calculate a meaningful effect size directly relating ratings to callback at the study level. Thus, we present a graphical representation in Fig 3B, with lines representing fitted linear models for each study (Table S7 in S3 File). For some categories, the relation between callback and PC1 is positive (e.g., nationality, which also samples a larger number—35—of categories). For most other categories, however, such as wealth, sexuality, and parenthood, there are both positive and negative slopes in different studies. Under the category of sexuality, slope signs in four studies were equally split between positive and negative, which is especially striking given the large range of ICCs across signals.

## Discussion and conclusion

Over the last few decades, many social scientists have used correspondence studies to document the disparities in outcomes that people experience in the labor market purely based on aspects of their social identity. Learning about these disparities is imperative for fostering fair and inclusive labor markets. Our study examined whether social perception predicts callback decisions in correspondence studies targeting the US and Canadian labor markets.

[13] found that 12 distinct subdimensions of social perception (including friendliness, sincerity, self-control, efficiency, and others) could be effectively condensed into the two factors of warmth and competence [27, 34]. Building upon this finding, we focused our investigation on participants' perceptions of others' warmth and competence based on attributes offered in the relevant correspondence studies, such as name, religion, or sexual orientation. We found that the perceived warmth and competence of individuals' names or attributes were highly correlated, leading us to use the first principal component (PC1) to measure favorable social perception. This component, reflecting both warmth and competence, explained about 70–81% of the variance in social perception ratings and was moderately linked to callback rates.

We found that in studies where names were varied to signal race, gender, and age, more favorable warmth and competence perception, based on names, positively predicted callback. However, for studies varying applicant characteristics such as sexuality and disability status, the effects of social perception on callback rates are ambiguous: some categories show a positive association between favorable social perception and callback rates, such as age and nationality, while others show a negative association. This result is unsurprising, given the small number of levels for some of the categories, and the effects observed for these categories are more susceptible to measurement issues like low inter-rater reliability (ICC).

The wide prediction interval for the positive correlation in our name analysis suggests that future studies might uncover negative correlations between positive ratings and callback rates. Our stringent selection criteria—restricting studies to those altering names to signify race and gender, conducted in North America, and offering raw data—resulted in a relatively small sample and excluded industry-specific variables. We found no publication bias in this sample. Moreover, our prediction approach adds variability, as it depends on perceptions from a group separate from the actual decision-makers, potentially contributing to the broad prediction interval.

Despite differences in the population that rated social perception and the employers making hiring decisions, there is a noteworthy predictive relationship between these ratings and callback rates, suggesting common cultural biases. The accuracy of these predictions could be even greater if the raters' demographics more closely matched those of the hiring decision-makers.

The reliability of categorical ratings in our study is measured by intraclass correlations (ICC). Certain social information categories, like military status and age, have shown low ICC, indicating raters' disagreement on the warmth and competence perceived in these groups. This disagreement suggests limitations in the predictive ability of social perception measurements for these categories. To enhance prediction, future studies could focus on gathering more ratings for categories with traditionally low ICC.

The validity of correspondence studies hinges on the subtle resume signaling of category membership being perceived by employers who read the resumes. Yet, monitoring how much attention these signals receive is challenging outside of laboratory or carefully designed field settings, which can track attention more directly [35, 36]. Signal effectiveness likely varies by category, affecting study outcomes. Our meta-analysis offers a view of the collective impact and variance of different signaling methods [35]. Future research should diversify the signals tested to understand their impact on labor market callbacks better.

The discussion of discrimination theories in our study pertains to foundational economic models, delineating primarily into taste-based [37] and statistical discrimination [38, 39] theories. Economists have refined "statistical discrimination" into "belief-based discrimination" [40, 41], where stereotypes informed by group averages fuel bias, closely resonating with our investigation's focus. Contrasting with these models, [2] suggest a "lexicographic search" pattern among hiring managers, which our Prolific setup reflects by presenting raters with names or categories to judge societal warmth and competence. However, since information like club memberships are typically listed later, this pattern may not always hold. Additionally, recruiters seem to review whole resumes but allocate less scrutiny to younger Black individuals' resumes [42]. Moreover, decision-makers exhibit prejudice that persists irrespective of new information about individuals [43], pointing to the tenacity of social perceptions and their predominant influence on callback decisions. Furthermore, employers' predispositions towards in-group members [44] underscores discrimination facilitated by either positive in-group or negative out-group stereotyping. Lastly, the institutional discrimination theory posits that discrimination's intensity is contextually determined [45], which is an area our study, constrained to the North American job market context, does not explore.

Our study's practical implications lie in harnessing the link between social perceptions and callback rates to refine recruitment practices. Decision support systems using this cognitive framework could help mitigate discrimination [46]. Furthermore, understanding discrimination via social perceptions facilitates generalization to underexplored stereotypes, crucial for protecting intersectional groups from bias. Perceptions associated with one group can inform on multiple intersecting identities [9]. Thus, our research advocates for a predictive model to anticipate labor market outcomes for intersectional groups, highlighting a direction for future bias mitigation efforts. Third, our study may contribute to the development of responsible AI by offering computer scientists insights into potential debiasing strategies proven effective in human decision-making, which can be translated into AI models. The literature on responsible AI most frequently measures bias by systematically prompting the model and analyzing the generated or retrieved image output. For example, the text prompt "CEO" will typically be more strongly associated with images of men than women [47, 48]. Most studies rely on sets of examples, e.g., various professions, to detect biases, thereby lacking a validated collection for comprehensively assessing biases. In contrast, an approach grounded in social perception moves beyond sets of examples, providing a broader framework. A few authors in the representation learning literature have seen value in this approach [49–52]. Specifically, [53] estimate bias in an image dataset by using text attributes of interest from social psychology and creating a set of text prompts. Their results reveal patterns of bias as well as noise in conventional bias measurements.

## Supporting information

**S1 File. Statistical definitions.**
(PDF)

**S2 File. Heterogeneity analysis.**
(PDF)

**S3 File. Subgroup analysis: Social perceptions across job types.**
(PDF)

**S4 File. Prolific sample.**
(PDF)

## Acknowledgments

We thank Beatrice Maule, Mario Paiva, and Alec Guthrie for their assistance with data collection and extraction research. Helpful comments were received from internal lab meetings at Caltech and from journal referees. We thank the authors who contributed data.

## Author Contributions

**Conceptualization:** Marcos Gallo, Carina I. Hausladen, Ming Hsu, Adrianna C. Jenkins, Vaida Ona, Colin F. Camerer.

**Data curation:** Marcos Gallo, Carina I. Hausladen.

**Formal analysis:** Marcos Gallo, Carina I. Hausladen.

**Funding acquisition:** Ming Hsu, Adrianna C. Jenkins, Colin F. Camerer.

**Investigation:** Marcos Gallo.

**Methodology:** Marcos Gallo, Carina I. Hausladen, Ming Hsu, Adrianna C. Jenkins, Vaida Ona, Colin F. Camerer.

**Project administration:** Ming Hsu, Adrianna C. Jenkins, Colin F. Camerer.

**Resources:** Ming Hsu, Adrianna C. Jenkins, Colin F. Camerer.

**Software:** Marcos Gallo, Carina I. Hausladen.

**Supervision:** Ming Hsu, Adrianna C. Jenkins, Colin F. Camerer.

**Validation:** Marcos Gallo, Ming Hsu, Adrianna C. Jenkins, Vaida Ona, Colin F. Camerer.

**Visualization:** Marcos Gallo, Carina I. Hausladen.

**Writing – original draft:** Marcos Gallo, Carina I. Hausladen.

**Writing – review & editing:** Marcos Gallo, Carina I. Hausladen, Adrianna C. Jenkins, Vaida Ona, Colin F. Camerer.

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
