## [Decision Letter · Decision Letter 0]

26 Feb 2024

PONE-D-24-01231Perceived warmth and competence predict callback rates in meta-analyzed North American labor market experimentsPLOS ONE

Dear Dr. Hausladen,

Thank you for submitting your manuscript to PLOS ONE. After careful consideration, we feel that it has merit but does not fully meet PLOS ONE’s publication criteria as it currently stands. Therefore, we invite you to submit a revised version of the manuscript that addresses the points raised during the review process.

We look forward to receiving your revised manuscript.

Kind regards,

Tobias Otterbring

Academic Editor

PLOS ONE

“NSF DRMS grants 1851879 (ACJ), 1851745 (CFC), 1851902 (MH), and a Tianqiao and Chrissy Chen Graduate Fellowship (MG)”

“This research was supported by NSF DRMS grants 1851879 (ACJ), 1851745 (CFC),

1851902 (MH), and a Tianqiao and Chrissy Chen Graduate Fellowship (MG). We thank

Beatrice Maule, Mario Paiva, and Alec Guthrie for their data collection and extraction

research assistance. Helpful comments were received from internal lab meetings.

The authors declare no conflict of interests.”

“NSF DRMS grants 1851879 (ACJ), 1851745 (CFC), 1851902 (MH), and a Tianqiao and Chrissy Chen Graduate Fellowship (MG)”

5. We notice that your supplementary files are included in the manuscript file. Please remove them and upload them with the file type 'Supporting Information'. Please ensure that each Supporting Information file has a legend listed in the manuscript after the references list.

Additional Editor Comments:

Dear authors,

Your paper was sent to multiple potential reviewers. Unfortunately, most of them declined the request to review this manuscript. Nevertheless, I have now received feedback from one reviewer with considerable knowledge and expertise in meta-analytic work and various theories from economics. The reviewer notes several positive aspects of your work. However, he/she also expresses certain concerns. The comments from the reviewer are attached below.

Based on the input from the reviewer and my own reading of the manuscript, I am willing to provide you with the opportunity to move this manuscript into a second round of reviews. Thus, I invite you to undertake a revision with a relatively clear path toward publication as long as you meticulously address all the feedback points from the reviewer and incorporate needed changes or edits to the manuscript.

Please address the comments from the reviewer in a very careful and sincere manner. If for some reason, you do not wish to address a given suggestion, highlight the reasoning in your response letter. The reviewer has made many insightful comments so please be very responsive.

Kind regards,

Tobias Otterbring

Handing Editor, PLOS One

Reviewers' comments:

Reviewer's Responses to Questions

**Comments to the Author**

1. Is the manuscript technically sound, and do the data support the conclusions?

Reviewer #1: Yes

2. Has the statistical analysis been performed appropriately and rigorously? 

Reviewer #1: Yes

3. Have the authors made all data underlying the findings in their manuscript fully available?

Reviewer #1: Yes

4. Is the manuscript presented in an intelligible fashion and written in standard English?

Reviewer #1: Yes

5. Review Comments to the Author

Reviewer #1: This paper reports the results of a study investigating how callback rates from correspondence studies investigating labor market discrimination might be predicted by the social signals of warmth and competence. The authors do so by meta-analyzing 21 published correspondence studies and collecting ratings of warmth and competence from a set of independent raters (Prolific participants). The authors find that social perceptions of warmth and competence predicts callback disparities for studies varying race and gender, but that the results are inconsistent for studies varying other categories such as sexuality or disability.

I would like to applaud the authors for their extensive and transparent reporting of their methods, which makes all analytical decisions and the corresponding results, concise and clear in their manuscript. In relation to this, it’s also great to see that the authors test the sensitivity of their analytical strategies by e.g. calculating various forms heterogeneity measures (instead of just relying on one type of analysis) to gain a fuller picture of the between- and within-study heterogeneity in the analyzed studies.

In general, I think the authors have done a very good job in clearly presenting their methods, results and a clear discussion of the theoretical and practical implications of these findings. That is, I think the paper is already in very good shape. However, throughout my review of the manuscript, I identified some issues that I outline below.

1. From the methods section, it is unclear what the authors’ justification is for using a random-effects model meta-analysis. I think it would be valuable to include a short (1-3 sentences) argument for why this model is the appropriate one to use. For instance, did all papers only include one study or did some include multiple? Did several papers originate from the same set of authors/universities/labs? Basically, are there any specific clusters in the data that would have made a mixed-effect model meta-analysis a more appropriate choice?

2. What is the rationale for the sample size of the Prolific raters? Also, was this a convenience (vs. nationally representative) Prolific sample and how was the sample quality-checked? Were there for instance any attention/bot/comprehension/quality- checks? Since the ICC’s are also an integral part of assessing the quality of the rating for the raters, I suggest the authors draw these results more to the front in the method section on p. 5.

3. I was also wondering what the rationale for using a Prolific sample is here? Would the authors have expected a different result if e.g. the raters had previous experience with hiring decisions?

4. On page 5 the authors report “…a marginally significant difference in competence between black and white (-11.52, p = .06)”. Considering the massive problems of replication and reproducibility across the behavioral science, I suggest the authors refrain from using such language.

5. On p. 6, the authors highlight the large prediction interval for the meta-analysis because of the substantial heterogeneity. I think the authors could address this result more in the discussion; for instance, could this result allude to that this literature could benefit from a more standardized approach of testing for labor market discrimination?

6. I was surprised to see that the authors do not test for publication bias in the meta-analysis. If substantial publication bias is present in the current literature, this will naturally influence how much we can trust the findings of the meta-analysis. I think the authors need to conduct analysis of publication bias and correct the estimates for publication-bias.

7. The exploratory analysis in the last paragraph of section 3 (page 8) only reports the correlation coefficients but not p-values and CI’s. I would encourage the authors to report these test statistics to allow the reader a more complete overview of these results.

8. Some signals, such as those for wealth and sexuality, yield vastly different ICC’s as noted by the authors in the first paragraph of section 4 (page. 8). Why might this be? Could it be a result of certain individual differences in e.g. political orientation etc.? I think it would be valuable if the authors further explore this to identify if there are any systematic influences for these differences. The authors outline this as point of discussion on page 10., but I think they could have elaborated more on this aspect.

9. Lastly, while I agree that the present work can be valuable for the development of responsible AI, I think the authors need to expand on this argument. Right now, this implication comes of as overstated and general.

Having mentioned these issues, I wish the authors all the best for their project and I am truly looking forward to reading their revised manuscript, either again for review or published.

6. PLOS authors have the option to publish the peer review history of their article (what does this mean?). If published, this will include your full peer review and any attached files.

Reviewer #1: **Yes: **Christian T. Elbaek

---

## [Decision Letter · Decision Letter 1]

17 May 2024

Perceived warmth and competence predict callback rates in meta-analyzed North American labor market experiments

PONE-D-24-01231R1

Dear Dr. Hausladen,

We’re pleased to inform you that your manuscript has been judged scientifically suitable for publication and will be formally accepted for publication once it meets all outstanding technical requirements.

Kind regards,

Tobias Otterbring

Academic Editor

PLOS ONE

Additional Editor Comments (optional):

Dear authors,

Based on the assessment of the expert reviewer and my own assessment of your manuscript, I am happy to inform you that my recommendation as the handling editor is to accept your paper for publication in PLOS One in its current form. Well done!

Kind regards,

Tobias Otterbring

Handling Editor

Reviewers' comments:

Reviewer's Responses to Questions

**Comments to the Author**

1. If the authors have adequately addressed your comments raised in a previous round of review and you feel that this manuscript is now acceptable for publication, you may indicate that here to bypass the “Comments to the Author” section, enter your conflict of interest statement in the “Confidential to Editor” section, and submit your "Accept" recommendation.

Reviewer #1: All comments have been addressed

2. Is the manuscript technically sound, and do the data support the conclusions?

Reviewer #1: Yes

3. Has the statistical analysis been performed appropriately and rigorously? 

Reviewer #1: Yes

4. Have the authors made all data underlying the findings in their manuscript fully available?

Reviewer #1: Yes

5. Is the manuscript presented in an intelligible fashion and written in standard English?

Reviewer #1: Yes

6. Review Comments to the Author

Reviewer #1: (No Response)

7. PLOS authors have the option to publish the peer review history of their article (what does this mean?). If published, this will include your full peer review and any attached files.

Reviewer #1: **Yes: **Christian T. Elbaek

---

## [Editor Report · Acceptance letter]

13 Jun 2024

PONE-D-24-01231R1 

PLOS ONE

Dear Dr. Hausladen, 

I'm pleased to inform you that your manuscript has been deemed suitable for publication in PLOS ONE. Congratulations! Your manuscript is now being handed over to our production team.

Kind regards, 

on behalf of

Professor Tobias Otterbring 

Academic Editor

PLOS ONE